# Automatic Instruction Data Selection for Large Language Models via Uncertainty-Aware Influence Maximization

## Abstract

Recent years have witnessed the prevalent integration of Large Language Models (LLMs) in various Web applications, such as search engines and recommender systems. As an emerging technique, instruction tuning aims to align pre-trained LLMs as capable chatbots that excel at following human instructions. Previous research indicates that selecting an appropriate subset of a large instruction dataset can enhance the capabilities of LLMs and reduce training costs. However, existing works tend to overlook external correlations between instruction examples during data selection process, which can introduce potential bias and lead to sub-optimal performance. To bridge this gap, we formalize this problem from graph influence maximization perspective and propose **Un**certainty-aware **i**nfluence **Max**imization (UniMax), a data selection framework that explicitly incorporates the complex inter-dependencies within instruction data. Specifically, we first define a latent instruction graph, treating each instruction example as a graph node and representing their implicit relations as graph edges. Instead of solely relying on heuristic metrics for graph construction, we develop a self-supervised graph learner to uncover the latent structure beyond surface-level feature correlations. After that, we propose an uncertainty-aware influence function to score each example on the instruction graph, allowing a simple greedy algorithm to select a valuable subset that embodies both high influence and uncertainty with an approximation guarantee. Extensive experiments on public datasets show that the proposed approach can significantly enhance model capabilities, underscoring the importance of exploiting data dependencies in instruction data selection.

## CCS Concepts

• **Do Not Use This Code → Generate the Correct Terms for Your Paper**; *Generate the Correct Terms for Your Paper*; Generate the Correct Terms for Your Paper; Generate the Correct Terms for Your Paper.

## Keywords

Do, Not, Us, This, Code, Put, the, Correct, Terms, for, Your, Paper

**ACM Reference Format:**
Anonymous Author(s). 2018. Automatic Instruction Data Selection for Large Language Models via Uncertainty-Aware Influence Maximization. In *Proceedings of Make sure to enter the correct conference title from your rights confirmation emai (Conference acronym 'XX)*. ACM, New York, NY, USA, 10 pages. https://doi.org/XXXXXXX.XXXXXXX

## 1 Introduction

In recent years, we are witnessing the widespread adoption of Large Language Models (LLMs) (*e.g.*, GPT-4o [1] and LLaMA [35]) in various Web-based applications and services, including search engines [48, 54], recommender systems [52, 55], and content generation tools [1]. Typically, LLMs are pre-trained on extensive text corpora from the Web [27], which learn universal representations that can be transferred to a wide range of language-related tasks. However, the pre-trained models often struggle with following human instructions. To unlock such capabilities, instruction tuning [26, 42, 51] has become a common practice, in which pre-trained LLMs are further refined to adeptly follow instructions using massive instruction-response pairs. Such a process not only enables non-experts to interact with LLMs in a controllable way, but also facilitates model generalization to unseen tasks.

Considerable efforts have been made to scale up instruction datasets by manually or automatically creating more instruction-response pairs [9, 33, 42]. Nevertheless, the prevalence of large-scale instruction datasets also poses significant computational challenges for training LLMs, particularly for small organizations and companies with limited resources. More dramatically, the computational cost continually increases under specific scenarios that require fine-tuning the LLMs multiple times, *e.g.*, continual learning and model selection, resulting in slower model iteration and experimentation. On the other hand, there is increasing evidence that data quality is more critical than quantity for improving instruction tuning. The superficial alignment hypothesis, introduced by LIMA [53], suggests that all the necessary knowledge in LLMs is already acquired during pre-training, and a small number of valuable instruction examples are sufficient for teaching them to follow specific response styles or formats. LIMA shows that fine-tuning LLMs with only 1,000 high-quality instruction examples can induce remarkably strong generalization capabilities. Therefore, data selection is worth exploring in instruction tuning, as it not only improves the instruction-following skills but also speeds up model development and deployment in online scenarios.

Manually selecting instruction data is a laborious, costly, and error-prone process. As a result, it is desirable to develop automated methods that can economically and efficiently find the most informative data examples. Existing approaches primarily focus on evaluating individual data points through either pre-defined indicators [21] or powerful reference LLMs [6, 25], *e.g.*, ChatGPT. For instance, AlpaGasus [6] calculates the quality score of each instruction example by directly prompting ChatGPT and filters out low-quality data. IFD [21] selects instruction data using a prior metric that quantifies the discrepancy between model output and expected response. In addition, another branch of methods [23, 46] have been

developed recently to select data relevant to specific model capabilities. The most representative work is LESS [46], which identifies valuable instruction data by using gradient features to measure the similarity with a handful of examples that reflect a target capability.

While the aforementioned methods have achieved decent results, they mainly rely on internal information within the individual instruction example, overlooking potential correlations with other examples. In fact, instruction data often exhibit complex interdependence nature, *e.g.*, two semantically different examples may share similar reasoning process [46]. Unlike pre-training stage, the primary focus of instruction tuning is to learn the response style rather than semantic knowledge, failing to consider inter-example correlations might introduce similar, redundant, or even counterproductive examples, leading to sub-optimal performance. How to effectively incorporate such external information for instruction data selection remains an open question.

To address the above limitation, we propose **Un**certainty-aware **i**nfluence **Max**imization (UniMax), a relation-powered data selection framework for instruction tuning. Specifically, we first formulate instruction data selection as an influence maximization problem, which aims at finding a data subset in a graph connecting instruction examples that maximize the spread of influence. We start with representing the target instruction dataset as a latent graph, where each node is an instruction example and edges indicate implicit relations between them. However, a critical challenge is that due to the heavy computational overhead, directly applying LLMs to quantify correlations among arbitrary instruction pairs can be expensive and time-consuming. To address this issue, we devise a small self-supervised instruction graph learner as a proxy for uncovering implicit data dependencies. The graph learner can efficiently distill adaptive hidden graph structure among all the instruction embeddings generated from a pre-trained LLM, under the guidance of the instructions themselves.

Then, with such a graph structure, the data selection can be regarded as solving influence maximization problem on the instruction graph. Nevertheless, simply maximizing influence within graph neglects the inherent uncertainty of each example with respect to target model, which introduces systematic bias to the selected set and thus undermine the effectiveness of instruction tuning. To this end, we further propose uncertainty-aware influence function to balance the effect of influence and uncertainty. Concretely, we explicitly incorporate the instruction-following uncertainty score to amplify the influence of nodes with high uncertainty scores while restricting the influence of low uncertainty nodes. Moreover, the function also satisfies properties of monotonicity and submodularity, allowing a fast greedy selection algorithm to find a near-optimal solution with theoretical guarantee.

Our contributions are summarized as follows: (1) We reframe instruction data selection as an influence maximization problem, which aims to select an instruction subset that maximizes the number of instruction examples that are influenced. (2) We develop a self-supervised graph learner to capture the latent instruction graph structure. The complexity scales linearly with the number of instruction examples. (3) We further propose a new metric for instruction data selection by unifying data influence and uncertainty into an uncertainty-aware influence maximization framework. (4) Extensive experimental results show significant improvements of the proposed approach over instruction data selection baselines. Notably, by selecting 10% of the data, our method often yields on-par or even better results than training on the full dataset, and the performance remains robust across different model scales.

## 2 Preliminaries

In this section, we first briefly introduce the backgrounds of instruction tuning and influence maximization. Then, we formally define the instruction selection problem.

### 2.1 Instruction Tuning

Instruction tuning refers to the process of fine-tuning pre-trained LLMs using a set of instruction-response pairs [40], *i.e.*, instruction examples. This is an essential step for adapting LLMs to unseen tasks and scenarios, as well as for stimulating the instruction-following capabilities of LLMs. Formally, an instruction example can be defined as follows.

DEFINITION 1. ***Instruction example***. *An instruction example is formatted as a chatbot-style example consisting of interactions $\{(x,y)\}$[1] between the user and the LLM, where $x$ is a user prompt and $y$ denotes expected response.*

Given an instruction set $\{(x_i, y_i)\}_{i=1}^N$, where $N$ is the number of examples, instruction tuning trains the LLMs in a supervised learning manner, *i.e.*, only computing the loss on the tokens (*i.e.*, words) belonging to the response $y$, defined as

$$m_j = \begin{cases} 1, & \text{if } t_j \in y \\ 0, & \text{otherwise} \end{cases} \quad (1)$$

$$\mathcal{L} = -\sum_j m_j \log p_\theta\left(t_j \mid t_{<j}\right), \quad (2)$$

where $t_j$ denotes the $j$-th token that can be either from user prompt $x$ or response $y$, $p_\theta$ is the probability of generating $t_j$ given $t_{<j}$, and $m_j = 1$ if $t_j \in y$ otherwise $m_j = 0$. In practice, the success of instruction tuning is powered by two critical components: (1) a powerful pre-trained LLM model, and (2) a diverse and representative instruction dataset.

### 2.2 Influence Maximization

The goal of Influence Maximization (IM) is to find a set of $K$ users (*i.e.*, seed set) that result in the highest spread of influence to other users in a given social network [18]. Formally, suppose we have a social network represented as a graph $\mathcal{G} = (\mathcal{V}, \mathcal{E})$, where $\mathcal{V}$ and $\mathcal{E}$ are the set of nodes and edges, respectively. The goal of IM is to select $K$ nodes from the graph $\mathcal{G}$ as the seed set $S$ to maximize influence in information diffusion, defined as

$$S^* = \arg\max_S \sigma(S), s.t. S \in \mathcal{V}, |S| = K \quad (3)$$

where $\sigma(\cdot)$ is the influence function that estimates the expected number of nodes influenced by $S$, following specific diffusion models like the independent cascade model [22]. The diffusion model typically involves two steps. First, it marks the status of nodes in seed set as active and other nodes as inactive. Then, it allows active

---

[1]For the sake of simplicity, we assume multi-turn instructions as single-turn instructions.

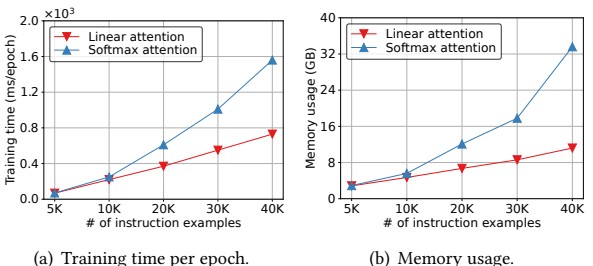

(a) Training time per epoch.          (b) Memory usage.

**Figure 1: Efficiency comparison between linear attention and Softmax attention.**

nodes to activate their neighbors through information spread until no new nodes can be activated. Although the IM problem is NP-hard in general, the optimal solution $\sigma(S^*)$ can be approximated with theoretical guarantee. In particular, if $\sigma(\cdot)$ is monotone and submodular, a simple greedy algorithm can return a node set $\hat{S}$ such that $\sigma(\hat{S}) \geq (1 - 1/e)\sigma(S^*)$.

### 2.3 Problem Statement

In this paper, we study the instruction selection problem, *i.e.*, identifying the most informative instruction examples for instruction tuning, which is defined below.

PROBLEM STATEMENT 1. *Given a large pool of instruction examples* $\mathcal{V} = \{(x_i, y_i)\}_{i=1}^N$ *and a fixed data budget* $K < N$*, the goal is to select a subset* $\mathcal{B}^*$ *of size* $K$ *from the full set* $\mathcal{V}$ *to maximize the instruction tuning effectiveness:*

$$\mathcal{B}^* = \arg\max_{\mathcal{B}} E(\mathcal{M}_\theta, \mathcal{B}), s.t. \mathcal{B} \in \mathcal{V}, |\mathcal{B}| = K, \quad (4)$$

*where* $E(\mathcal{M}_\theta, \mathcal{B})$ *is the performance of the language model* $\mathcal{M}_\theta(\cdot)$ *trained under the supervision of instruction set* $\mathcal{B}$*.*

## 3 Methodology

This section elaborates the proposed UniMax framework. Unlike prior works, we explicitly incorporate relational information of instruction examples into the data selection process. We will introduce each module of our method in detail below.

### 3.1 Overall Pipeline

Figure 2 shows an overview of the proposed approach, which consists of three major tasks: (1) instruction graph structure learning, (2) uncertainty-aware influence estimation, and (3) instruction data selection. To be specific, in the first task, we encode all instruction examples into embedding vectors via a pre-trained LLM and learn the latent graph structure among instruction embeddings with contrastive learning. In the second task, we quantify the value of each instruction example by jointly considering its influence on the graph and the inherent instruction-following uncertainty. In the third task, we adopt a simple greedy algorithm to select the most valuable data points by maximizing the uncertainty-aware influence within the latent graph.

### 3.2 Instruction Graph Structure Learning

Consider a set of instruction examples $\{(x_i, y_i)\}_{i=1}^N$, we measure their correlations using a latent graph $\mathcal{G} = (\mathcal{V}, \mathcal{E})$, where each instruction is represented as a node $v \in \mathcal{V}$ on graph and edge $e_{i,j} \in \mathcal{E}$ captures relation between node $i$ and $j$. To learn the graph, we first concatenate the prompt $x_i$ and response $y_i$ to obtain the complete example $x_i||y_i$, and then encode all examples into embedding vectors through a pre-trained LLM $\mathcal{M}_\theta(\cdot)$, *i.e.*, $\mathbf{x}_i = \mathcal{M}_\theta(x_i||y_i)$. Specifically, we extract the last token embeddings in the final layer as instruction embeddings, which will be utilized in subsequent graph construction.

We regard the graph $\mathcal{G}$ as a fully-connected graph and leverage a single layer self-attention operator to calculate edge weight between instruction node $i$ and $j$, denoted as

$$e_{i,j} = \mathbf{q}_i^\top \mathbf{k}_j, \quad (5)$$

$$\mathbf{q}_i = \mathbf{W}_q \mathbf{x}_i, \mathbf{k}_j = \mathbf{W}_k \mathbf{x}_j, \quad (6)$$

where $\mathbf{W}_q$ and $\mathbf{W}_k$ denote learnable projection matrices. We further normalize $e_{i,j}$ across all choice of $j$ via Softmax operator and compute the output representation for node $i$ as follows

$$\alpha_{i,j} = \frac{\exp(e_{i,j})}{\sum_{k=1}^N \exp(e_{i,k})}, \quad (7)$$

$$\mathbf{x}_i' = \sum_{j=1}^N \alpha_{i,j} \mathbf{W}_v \mathbf{x}_j, \quad (8)$$

where $\mathbf{W}_v$ is a learnable matrix. To reduce the quadratic complexity of self-attention computation and boost the graph structure learning efficiency, we replace the Softmax attention operator with linear attention [5, 45], defined as

$$\mathbf{x}_i' = \sum_{j=1}^N \frac{(\mathbf{q}_i')^T \mathbf{k}_j'}{\sum_{r=1}^N (\mathbf{q}_i')^T \mathbf{k}_r'} \mathbf{v}_i = \frac{(\mathbf{q}_i')^T \sum_{j=1}^N \mathbf{k}_j' \mathbf{v}_j}{(\mathbf{q}_i')^T \sum_{r=1}^N \mathbf{k}_r'}, \quad (9)$$

where $\mathbf{q}_i' = \sigma(\mathbf{q}_i)/\|\sigma(\mathbf{q}_i)\|_2$, $\mathbf{k}_i' = \sigma(\mathbf{k}_i)/\|\sigma(\mathbf{k}_i)\|_2$, where $\sigma(\cdot)$ represents ReLU activation function. With the above operator, we only need to compute $\sum_{j=1}^N \mathbf{k}_j' \mathbf{v}_j$ and $\sum_{r=1}^N \mathbf{k}_r'$ once and reuse them for each query, thus reducing the computational and storage complexity to linearity. Figure 1 shows that linear attention is faster and has a lower memory footprint than Softmax attention, which is more suitable for graph structure learning.

After obtaining the aggregated representation $\mathbf{x}_i'$ for each node, we leverage the input feature vector $\mathbf{x}_i$ itself as supervision signal to guide the graph structure learning process. However, $\mathbf{x}_i'$ has already incorporated the information of centered nodes through the self-loop propagation, which may lead to potential information leakage. Thus, we eliminate the centered node information through $\mathbf{x}_{i,h} = \mathbf{x}_i' - \frac{1}{d_i} \cdot \mathbf{v}_i$, where $d_i = \frac{(\mathbf{q}_i')^T \mathbf{k}_i'}{(\mathbf{q}_i')^T \sum_{r=1}^N \mathbf{k}_r'}$ is the self-loop edge weight. Afterward, we maximize mutual information between instruction embedding $\mathbf{x}_i$ and aggregated neighborhood embedding $\mathbf{x}_{i,h}$ via contrastive learning [8]. In specific, we map $\mathbf{x}_i$ and $\mathbf{x}_{i,h}$ to the representation space where contrastive loss is applied

$$\mathbf{z}_i = \text{MLP}_1(\mathbf{x}_i), \mathbf{z}_{i,h} = \text{MLP}_2(\mathbf{x}_{i,h}), \quad (10)$$

where $\text{MLP}_1(\cdot)$ and $\text{MLP}_2(\cdot)$ are Multi-Layer Perceptrons (MLPs). Then, we randomly sample a minibatch of $m$ instruction nodes and

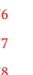
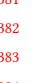

**Figure 2: An overview of the proposed UniMax framework.**

utilize contrastive loss to learn the graph structure by maximizing the agreement between $\mathbf{z}_i$ and $\mathbf{z}_{i,h}$ of the same node $i$

$$\mathcal{L} = -\log \frac{\exp(\text{sim}(\mathbf{z}_i, \mathbf{z}_{i,h})/\tau)}{\sum_{k=1}^{m} \exp(\text{sim}(\mathbf{z}_i, \mathbf{z}_{k,h})/\tau)}, \qquad (11)$$

where $\tau$ stands for temperature parameter and $\text{sim}(\cdot)$ is the cosine similarity function. Finally, we can derive the adaptive graph adjacency matrix by $\mathbf{A} = f_{sp}(\mathbf{Q}'(\mathbf{K}')^\top) + f_{sp}(\mathbf{X}\mathbf{X}^\top)$, where the $i$-th row of $\mathbf{Q}'$, $\mathbf{K}'$, and $\mathbf{X}$ are $\mathbf{q}'_i$, $\mathbf{k}'_i$, and $\mathbf{x}_i$ respectively, $f_{sp}(\cdot)$ sparsifies matrix by a threshold $s$.

### 3.3 Uncertainty-Aware Influence Estimation

Building upon the learned graph structure, we quantify the influence of node $v_i$ on $v_j$ by using the sum of probabilities of all possible paths with a length of $l$ from $v_i$ to $v_j$ on the graph [37, 47, 49]. Formally, we define the node influence score of node $i$ on $j$ as $I(v_j, v_i, l) = \sum_{\mathcal{P}_l^{v_i \to v_j}} \prod_k A_{v_k, v_{k+1}}$, where $\mathcal{P}_l^{v_i \to v_j}$ is a path from $v_i$ to $v_j$ and $A_{v_k, v_{k+1}}$ denotes the edge weight of $\mathbf{A}$.

However, simply selecting influential examples based on scoring function $I(v_j, v_i; l)$ may lead to a subset with skewed distribution over instruction-following uncertainty, which could degrade the performance of instruction tuning. As a result, we develop a new metrics to account for the influence and uncertainty of each data instance simultaneously. Given a user prompt $x$, we can measure the degree to which the output of pre-trained LLM $\mathcal{M}_\theta(\cdot)$ matches

**Algorithm 1:** Instruction Data Selection.

---

**Input:** Instruction dataset $\mathcal{V} = \{(x_i, y_i)\}_{i=1}^N$, data budget $K$, influential path length $l$, pre-trained LLM $\mathcal{M}_\theta(\cdot)$.
**Output:** Selected instruction set $\mathcal{B}$
$\mathcal{B} = \emptyset$;
Learn latent instruction graph structure $\mathcal{G}$ by Equation 11;
Calculate uncertainty score $d_i$ for each instruction example by Equation 13;
**for** $k = 1, 2, \ldots, K$ **do**
    Select the instruction example
    $v^* = \arg\max_{v \in \mathcal{V} \setminus \mathcal{B}} (\Phi(\mathcal{B} \cup \{v\}) - \Phi(\mathcal{B}))$;
    Update the instruction set $\mathcal{B} \leftarrow \mathcal{B} \cup \{v^*\}$;
**end**
**return** $\mathcal{B}$

---

the corresponding response $y$ by the following function

$$d(y \mid x) = -\frac{1}{T} \sum_{i=1}^{T} \log p_\theta(t_i^y \mid x, t_1^y, t_2^y, \ldots, t_{i-1}^y), \qquad (12)$$

where $t_i^y$ indicates the $i$-th token in the response $y$, and $p_\theta(\cdot)$ calculates the probability of generating the next token based on the user prompt and its preceding tokens. A higher $d(y \mid x)$ implies that the instruction is more challenging for the target model to follow. Subsequently, we normalize $d(y \mid x)$ and introduce the

**Table 1: Details of instruction datasets used in our experiments, which is from Tulu [40].**

| Datasets | Sourced from | # Instance | Instruction Length | Response Length |
|---|---|---|---|---|
| Flan-V2 | NLP datasets and human-written instructions | 100,000 | 355.7 | 31.2 |
| CoT | NLP datasets and human-written CoTs | 100,000 | 266.0 | 53.2 |
| Code-Alpaca | Generated from Davinci-003 | 20,022 | 35.6 | 67.8 |

uncertainty-aware influence score $I_d$ as follows

$$d_i = \frac{2d(y \mid x)}{d(x) + d(y)}, \tag{13}$$

$$I_d(v_j, v_i, k) = d_i I(v_j, v_i, l), \tag{14}$$

where $d_i$ represents the normalized uncertainty score of instruction node $i$. Intuitively, $I_d(v_j, v_i, k)$ demonstrates that the impact of node $v_i$ on $v_j$ is more significant if (1) The instruction is harder to follow and thus has higher uncertainty score $d_i$; (2) There are more influential paths with high probabilities from node $v_i$ to $v_j$ and thus leads to larger $I(v_j, v_i, l)$. Given a set of instruction nodes $\mathcal{B} \in \mathcal{V}$, we define the node set activated by $\mathcal{B}$ as follows

$$\Phi(\mathcal{B}) = |\{v | I_d(v, \mathcal{B}, l) > \epsilon, v \in \mathcal{V}\}|, \tag{15}$$

where $I_d(v, \mathcal{B}, l) = \max_{v_i \in \mathcal{B}} I_d(v, v_i, l)$ is the maximum uncertainty-aware influence score of $\mathcal{B}$ on node $v$, and $\epsilon$ is a hyper-parameter filtering weakly influenced nodes.

### 3.4  Instruction Data Selection

Finally, we select instructional examples by optimizing the following uncertainty-aware influence maximization objective

$$\max_{\mathcal{B}} \Phi(\mathcal{B}), s.t. \, \mathcal{B} \subseteq \mathcal{V}, \, |\mathcal{B}| = K. \tag{16}$$

By jointly considering both the magnitude of influence and the inherent uncertainty, our goal is to identify a node subset of size $K$ such that the maximum number of activated nodes can be attained.

**Greedy instruction selection.** As discussed in Section 2.2, the optimal solution of influence maximization can be approximated by greedy algorithm if $\Phi(\mathcal{B})$ has two properties: monotonicity and submodularity [18, 22], i.e., $\forall \mathcal{B} \subseteq \mathcal{B}' \subseteq \mathcal{V}, v \in \mathcal{V} \backslash \mathcal{B}', \Phi(\mathcal{B}') \geq \Phi(\mathcal{B})$ and $\Phi(\mathcal{B} \cup \{v\}) - \Phi(\mathcal{B}) \geq \Phi(\mathcal{B}' \cup \{v\}) - \Phi(\mathcal{B}')$. Thus, we validate the properties of $\Phi(\mathcal{B})$ for influence maximization below.

THEOREM 1.  *The uncertainty-aware influence function $\Phi(\mathcal{B})$ is monotone and submodular.*

Proof of Theorem 1 is in Appendix A. Given such a function, we can apply a simple greedy algorithm for data selection in influence maximization with an approximation ratio of $1 - 1/e$, i.e., $\Phi(\hat{\mathcal{B}}) \geq (1 - 1/e)\Phi(\mathcal{B}^*)$. Specifically, we first initialize an empty node set $\mathcal{B}$. Then, we iteratively select the node generating the maximum marginal gain $\Phi(\mathcal{B} \cup \{v\}) - \Phi(\mathcal{B})$ and add it into $\mathcal{B}$. This process is repeated until there are $K$ nodes in $\mathcal{B}$. The overall pipeline is illustrated in Algorithm 1.

**Scaling to large instruction datasets.** Our approach exhibits high efficiency and scalability, as it enables instruction graph structure learning with linear complexity. For larger instruction datasets that cannot be loaded into a single GPU, we can leverage the data partitioning strategy. The core idea is to divide the input dataset into multiple clusters, which largely reduces the computational cost for subsequent graph structure learning and data selection. For example, we can use K-Means clustering [3] for data partitioning. Moreover, due to the linear complexity of graph structure learning, we can utilize large cluster size to preserve critical instruction connectivity as much as possible.

### 3.5  Experimental Setup

**Datasets**. We closely follow Tulu [39] and construct three representative instruction datasets for our experiments. Specifically, we use the following datasets: (1) Flan V2 consists of various NLP tasks created from existing datasets or written by humans [10], (2) CoT is a collection of instructions annotated by using chain-of-thoughts [43], (3) Code-Alpaca is an instruction dataset generated from GPT-4 using the Alpaca dataset [34] as inputs. These datasets cover different style of instructions ranging from synthetic, manually curated to distilled from commercial LLMs. We unify the diverse instructions to chatbot-style training examples with the formatting strategy of Tulu [39]. The details of data statistics are provided in Table 1.

**Evaluation**. As different instruction datasets reflect distinct capabilities, we choose MMLU [16], TyDiQA [11], GSM [12], BBH [32], and Codex-Eval (*i.e.*, HumanEval) [7] for model evaluation, which cover multi-faceted capabilities including factual knowledge, reasoning, multilinguality, and coding. Specifically, we evaluate the fine-tuning performance of Flan-V2 dataset on MMLU and TyDiQA. MMLU contains a set of multiple-choice questions across 57 subjects spanning from STEM, humanity to social science, which can be utilized for factual knowledge evaluation. TyDiQA is a question answering benchmark consisting of 11 typologically diverse languages for testing the model's multilingual capability. For CoT dataset, we leverage GSM and BBH, two commonly adopted benchmarks for measuring the mathematical and general reasoning capabilities. Besides, we also evaluate the models trained on Code-Alpaca dataset through Codex-Eval. We follow the evaluation pipeline of previous study [39] to test the model performance on target dataset. More evaluation details can be found in Appendix B.

**Baselines**. We compare our approach with the following baselines: (1) Random selection uniformly sample instruction examples from the dataset for instruction tuning; (2) Perplexity measures generated response perplexity and selects instruction data with high perplexity; (3) K-Center-Greedy (KCG) [30] algorithm selects instruction data by iteratively choose the data point farthest from the current set; (4) IFD [21] first quantifies the discrepancy between model output and desired response by using an entropy-based metric and then selects the most difficult data samples; (5) DEITA [25] is a score-first, diversity-aware instruction selection approach that identifies the most valuable data by jointly considering data complexity, quality, and diversity. Notably, we do not compare with

**Table 2: Comparison of our approach with baseline methods on LLaMA-2-7B and LLaMA-2-13B. Full indicates training on full dataset, and otherwise we select 10% of data using different instruction data selection methods. Bold numbers represents the best-performing data subset.**

| Training Datasets | Flan V2 | | | CoT | | | Code-Alpaca |
|---|---|---|---|---|---|---|---|
| Methods | MMLU | TydiQA | Average | GSM | BBH | Average | Codex-Eval |
| **Instruction tuning on LLaMA 7B** | | | | | | | |
| Full (100%) | 48.0 | 50.4 | 49.2 | 33.0 | 36.4 | 34.7 | 32.6 |
| Random | 42.1 | 47.4 | 44.8 | 22.5 | 39.4 | 31.0 | 25.5 |
| Perplexity | 40.4 | 41.1 | 40.8 | 7.5 | 34.4 | 20.9 | 15.3 |
| KCG | 31.9 | 49.2 | 40.1 | 14.0 | 37.3 | 25.7 | 28.6 |
| IFD | 44.6 | 39.9 | 42.3 | 22.0 | 31.6 | 26.8 | 26.8 |
| DEITA | 45.9 | 48.4 | 47.2 | 26.5 | 36.7 | 31.6 | 30.1 |
| Ours | **47.3** | **51.2** | **49.3** | **28.5** | **39.9** | **34.2** | **31.3** |
| **Instruction tuning on LLaMA 13B** | | | | | | | |
| Full (100%) | 51.8 | 53.2 | 52.5 | 47.5 | 46.2 | 46.7 | 38.3 |
| Random | 48.1 | 51.9 | 50.0 | 31.0 | 47.1 | 39.1 | 35.4 |
| Perplexity | 52.5 | 45.9 | 49.2 | 12.5 | 43.9 | 28.2 | 16.1 |
| KCG | 47 | 45.3 | 46.2 | 21.5 | **50.1** | 35.8 | 35.2 |
| IFD | 51.1 | 46.4 | 48.8 | 37 | 45.7 | 41.4 | 38.2 |
| DEITA | 50.5 | 52.1 | 51.3 | 36 | 46.3 | 41.2 | 39.6 |
| Ours | **53.3** | **54.6** | **54.0** | **39.5** | 47.7 | **43.6** | **41.7** |

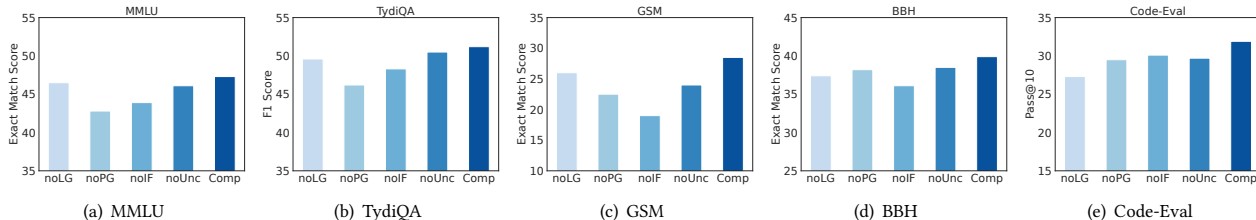

Figure 3: Ablation study on LLaMA-2-7b.

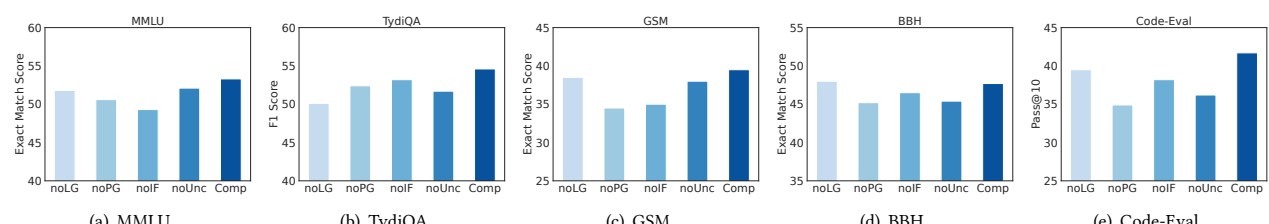

Figure 4: Ablation study on LLaMA-2-13b.

targeted data selection methods, such as LESS [46], as they highly rely on small reference set reflecting target capability, while our approach is non-targeted.

**Implementation details**. In this work, we choose two base models, *i.e.*, LLaMA-2-7B and LLaMA-2-13B, for our experiments. Concretely, we leverage a pre-trained LLaMA-2-7B model $\mathcal{M}_\theta$ to generate embeddings and uncertainty scores for all instruction samples in the original dataset. Afterward, we train a one-layer graph

learner to learn the dependencies from instruction embeddings. The number of hidden units in the graph learner is set to 2048, and we randomly sample $m = 8,192$ instruction nodes to calculate the contrastive loss in each training iteration. In uncertainty-aware influence scoring, we fix the path length $l$ to 1 for computing the influence score among nodes. The threshold $s$ and $\epsilon$ are set to 0.95 and 0.1. Finally, based on the selected data, we fully fine-tune the LLaMA-2-7B and LLaMA-2-13B models. The learning rate and batch

size are set to $2e − 5$ and 128, respectively. Besides, we also fix the maximum input sequence length to 2048, truncating samples if necessary. We conduct all experiments on a GPU cluster, where each computing node is equipped with 8 NVIDIA A100 GPUs, along with 71 TB disk storage and Intel (R) Xeon (R) Platium 8352Y 128-Core Processor. During training, we exploit the ZeRO optimizer [28] and DeepSpeed library [29] for large-scale model fine-tuning.

## 3.6 Performance Comparison

Table 1 shows the overall performance of our selection approach and all the baselines with respect to fine-tuned LLaMA-2-7B and LLaMA-2-13B on five evaluation datasets. We report the exact match score on MMLU, GSM, and BBH using 0, 8, and 3 few-shot in-context examples, respectively. For TydiQA, we compute the 1-shot F1 score across all multilingual questions under the Gold Passage (GP) setting, *i.e.*, the reference answer is provided to the model. For Code-Eval, we report the pass@10 results with a temperature of 0.8 to measure the correctness of model output. We summarize the key observations as follows.

Overall, our method consistently outperforms baseline methods across different model scales and evaluation datasets, indicating the effectiveness of selecting both influential and difficult examples for instruction tuning. Moreover, the proposed approach performs comparable or even surpasses the results achieved by training with the entire dataset. This implies that the full dataset may encompass low-quality data samples, which adversely affect the instruction tuning process. Our approach is able to identify the most valuable data while filtering negative samples, resulting in improved performance. Additionally, going deep into the baselines, we have the following findings: (1) perplexity-based selection strategy has the worst performance, which suggests that perplexity may not be a reliable metric for instruction data selection. (2) KGC and IFD outperform Perplexity by a large margin, underscoring the importance of data diversity and difficulty for instruction tuning, respectively. (3) Random selection selects data uniformly and achieves surprisingly good performance. The possible reason is that each example has an equal chance of being chosen, thereby maintaining representativeness of the entire dataset, whereas baselines like Perplexity and IDF usually leads to a subset that is biased towards a specific group of data samples. (4) DEITA attains the best performance among these baselines, demonstrating the advantage of jointly considering multi-dimensional factors, namely quality, complexity, and diversity, for data assessment. However, DEITA neglects fine-grained and latent dependencies within instruction data, which is also essential for data selection.

## 3.7 Ablation Studies

In this section, we conduct an ablation study to verify the effectiveness of each component. Specifically, we evaluate the following four variants: (1) *noLG* replaces the learned graph by pre-defined graph built with cosine similarity, (2) *noPG* only utilizes the graph structure learned via contrastive learning (*i.e.*, removing the second term in $\mathbf{A} = f_{sp}(\mathbf{Q}'(\mathbf{K}')^{\top}) + f_{sp}(\mathbf{X}\mathbf{X}^{\top})$), (3) *noIF* removes influence score for data selection, (4) *noUnc* excludes uncertainty score in uncertainty-aware scoring function, (5) *Complete* represents the original selection approach.

The results are reported in Figure 3 and 4. First, there is a performance degradation when using pre-defined graph for data selection. The possible reason is that pre-defined graph fails to capture complicated relationships between instruction examples, resulting in selection bias. Second, only leverage the learned graph structure without prior knowledge hinder the performance of instruction tuning. This is because the geometric prior of instruction embeddings also contains critical relational information, which can complement the purely learned graph. Third, the performance will decrease in all five datasets if the influence score is neglected in data selection, which demonstrates the effectiveness of maximizing the influence of selected instruction set. Finally, by adding uncertainty score to the influence function, we can achieve on-par or better performance, which further emphasizes the importance of uncertain examples. Besides, the data influence has more substantial impact on model performance than example uncertainty, as there is a more evident performance drop with the removal of influence score.

## 3.8 Parameter Sensitivity

Finally, we evaluate the impact of hyper-parameters on the performance of instruction tuning. Specifically, we examine the impact of the data budget $K$, the length of influential path $l$, and the activation threshold $\epsilon$, three crucial hyper-parameters that relate to our approach. We conduct analysis using LLaMA-2-13B. The results on LLaMA-2-7B are consistent with LLaMA-2-13B. All other hyper-parameters are set to their default values when evaluating the target one.

First, we adjust the data budget $K$ from 1% of data to 100% of data. The results across five evaluation datasets are presented in 5(a), where we find that in most cases, training with 10% of the data achieves acceptable balance between performance and efficiency. Besides, if researchers are more interested in improving model capabilities, we suggest searching an optimal value of data budget on a held-out validation dataset.

Then, we study the effect of influential path $l$ by increasing $l$ from 1 to 4. The results are reported in 5(b). We can see that the performance remain stable between 1 and 3; however, performance degrades when $l$ is further increased from 3 to 4. The potential explanation is that larger $l$ may cover more noisy instruction nodes that are weakly influenced by the target node, leading to inaccurate data selection.

Finally, we vary the activation threshold $\epsilon$ from 0.05 to 0.25. The results are illustrated in 5(c). Overall, the best performance is obtained when $\epsilon = 0.1$. We observe a performance gain when increasing $\epsilon$ from 0.05 to 0.1, further increasing $\epsilon$ from 0.1 to 0.25 results in performance deterioration. This decline occurs because larger $\epsilon$ may weaken the influence of some valuable instruction nodes, making them hard to be selected during the influence maximization process.

## 4 Related Works

## 4.1 Instruction Tuning

Instruction tuning aims to refine the pre-trained LLMs on a collection of datasets expressed via natural language instructions [51]. Early studies [36, 42] leverage instruction tuning to improve zero-shot generalization performance. For example, FLAN [42] fine-tunes

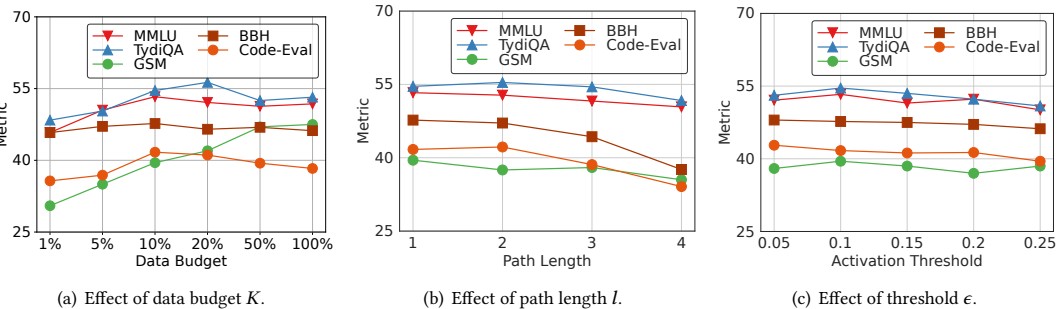

(a) Effect of data budget $K$.

(b) Effect of path length $l$.

(c) Effect of threshold $\epsilon$.

Figure 5: Parameter sensitivity analysis on LLaMA-2-13b.

a pretrained LLM on a broad range of NLP tasks described as instructions, which substantially enhances the model capability to perform tasks unseen during training. Conversely, InstructGPT [26] aims to align LLM with human preferences by fine-tuning with human-labeled instructions. Recently, a series of works have been done to improve the instruction following abilities of open-source LLMs [9, 33, 41, 53]. Among them, LIMA [53] suggests that data quality is more important than quantity during instruction tuning stage, which has sparked considerable efforts for instruction data selection [6, 20, 21, 23–25, 38]. To name a few, AlpaGasus [6] evaluates the quality of each instructional example through ChatGPT and filters out low-quality data. DEITA [25] introduces a multifaceted assessment strategy to select instruction data by simultaneously considering data complexity, quality, and diversity. However, existing approaches overlooks the complex dependencies within instruction data, such as semantically different examples that shares similar reasoning process, which largely limits the performance. In this work, we resolve this issue by formalizing instruction data selection as an influence maximization problem.

### 4.2 Data Selection

Data selection aims to improve data efficiency of machine learning models by choosing the most representative training samples, which has been widely used in various fields, such as computer vision [31] and graph mining [49, 50]. Existing studies mainly fall into two categories: active learning and core-set selection. Specifically, active learning focuses on identifying the most informative unlabeled samples for human labeling to maximize model performance [2, 14]. This process involves iteratively training a model on a labeled dataset and then selecting additional samples for labeling based on predefined metrics, such as the entropy of predicted distributions [17]. In contrast, the goal of core-set selection is to find a small subset that achieves performance comparable to the full dataset. In the past decade, many efforts are dedicated to developing data selection approaches, such as uncertainty sampling [15], greedy k-centers [30] and submodularity-based approaches [44]. Unfortunately, traditional data selection methods usually suffer from high computation cost when applying to deep learning, as they require retraining a deep model after each selection. Several studies [19, 30] focus on selecting samples in large batches to avoid

frequent model retraining, but they are still costly for large models. To address this issue, SVP [13] leverages a small proxy model, *i.e.*, the model with less hidden layers or fewer training epochs, to perform data selection more efficiently. In this paper, we explore instruction tuning data selection and devise a uncertainty-aware influence maximization framework to balance the effect of both data dependency and uncertainty.

## 5 Discussion and Future Work

While the proposed method demonstrate superior performance in our experiments, it still has room for improvement. We outline three major limitations of our approach as follows. First, we rely on an external graph structure learner to capture dependencies within instruction data. This approach might not accurately reflect the true data relationships from the perspective of the target model, potentially leading to bias towards spurious data correlations. Although we empirically demonstrate the effectiveness of this method, ensuring consistency between the graph structure learner and target model remains under-explored. Second, we leverage a data partitioning strategy to enhance the scalability for large instruction datasets, but data partitioning may induce information loss. To resolve this issue, additional efforts are required to explore advanced acceleration algorithms for influence maximization on extremely large graphs. Finally, our experiments are conducted on carefully curated open-source instruction datasets and do not explicitly consider the impact of noisy or incorrect instruction examples. We leave the exploration of instruction data selection under noisy setting for future work.

## 6 Conclusion

This paper studies instruction data selection from a new perspective of influence maximization. We propose an uncertainty-aware influence maximization framework for instruction data selection, which can exploit the interactions among data points by maximizing the number of examples influenced by selected instruction examples. To be specific, we first develop a self-supervised instruction graph learner as a proxy to capture dependencies among instructional examples. Then, we propose to unify example influence in the graph and its inherent uncertainty for instruction data selection. Finally, empirical results on public datasets verify the effectiveness of the proposed framework.

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

## A  Proof for Theorem 1

We provide more details about the monotonicity and submodularity properties for the proposed uncertainty-aware influence function $\Phi(\mathcal{B})$. Given any $\mathcal{B} \subseteq \mathcal{B}' \subseteq \mathcal{V}$, we have

$$\Phi(\mathcal{B}) = |\{v|I_d(v_i, \mathcal{B}, l) > \epsilon, v_i \in \mathcal{V}\}|, \tag{17}$$

$$\Phi(\mathcal{B}') = |\{v_i|I_d(v_i, \mathcal{B}', l) > \epsilon, v_i \in \mathcal{V}\}|. \tag{18}$$

Clearly, $\Phi(\mathcal{B}')$ tends to cover more activated nodes than $\Phi(\mathcal{B})$ when $\mathcal{B} \subseteq \mathcal{B}'$, *i.e.*, $\Phi(\mathcal{B}) \leq \Phi(\mathcal{B}')$. Thus, function $\Phi(\mathcal{B})$ is monotonically increasing.

In terms of submodularity, the function should satisfy $\Phi(\mathcal{B} \cup \{v\}) - \Phi(\mathcal{B}) \geq \Phi(\mathcal{B}' \cup \{v\}) - \Phi(\mathcal{B}')$. Here $\Delta = \Phi(\mathcal{B} \cup \{v\}) - \Phi(\mathcal{B})$ represents the additional influence gained by adding node $v$. Intuitively, it signifies the number of newly activated nodes, which can be written as

$$\Delta = |\{v|I_d(v_i, v, l) > \epsilon, I_d(v_i, \mathcal{B}, l) \leq \epsilon, v_i \in \mathcal{V}\}|. \tag{19}$$

Similarly, we have

$$\Delta' = |\{v|I_d(v_i, v, l) > \epsilon, I_d(v_i, \mathcal{B}', l) \leq \epsilon, v_i \in \mathcal{V}\}|. \tag{20}$$

By definition $I_d(v_i, \mathcal{B}, l) = \max_{v_j \in \mathcal{B}} I_d(v_i, v_j, l)$ we have

$$I_d(v_i, \mathcal{B}, l) \leq I_d(v_i, \mathcal{B}', l), \tag{21}$$

$$\{v|I_d(v_i, \mathcal{B}', l) \leq \epsilon\} \subseteq \{v|I_d(v_i, \mathcal{B}, l) \leq \epsilon\}. \tag{22}$$

As a result, we can obtain $\sigma(\mathcal{B} \cup \{v\}) - \sigma(\mathcal{B}) \geq \sigma(\mathcal{B}' \cup \{v\}) - \sigma(\mathcal{B}')$. In summary, the function $\sigma(\mathcal{B})$ is both monotone and submodular.

**Table 3: Efficiency analysis.**

| Module | Flan V2 | CoT | Code-Alpaca |
|---|---|---|---|
| GSL (Time) | 32 Min | 35 Min | 8 Min |
| GSL (Memory) | 8.8 GB | 10.5 GB | 4.7 GB |
| IM (Time) | 127 Min | 84 Min | 60 Min |
| IM (Memory) | - | - | - |

## B  Evaluation Setups

We elaborate on the evaluation setups utilized in our study. For the MMLU dataset, we use the official MMLU prompts and evaluation script, following the original MMLU configuration and employing a zero-shot setting to assess performance. For the GSM and BBH, we utilize Chain-of-Thought (CoT) prompting for model evaluation. Specifically, we adopt 8 and 3 few-shot examples for GSM and BBH respectively, where each CoT example include detailed reasoning steps. Since GSM answers are numerical numbers, we extract the last number from the model's response as the answer. To accelerate the evaluation process, we select a subset of 200 samples from the 1,319 test examples, which was shown to have performance similar to the full set previously [40]. For the BBH evaluation, we extract the first word following the phrase "so the answer is" and if this phrase was absent, we extract the entire response. For the TydiQA dataset, as described in the PaLM 2 technical report [4], we evaluate the model's ability to answer multilingual questions given the gold paragraph containing the answer (GoldP/GP). We use one in-context example to help the model get familiar with the answer format. Additionally, we employ the HumanEval dataset from the Codex paper [7] to evaluate the model's programming capabilities. This dataset comprises 164 programming problems where the model is prompted to complete Python functions based on their docstrings. Consistent with the original paper, we compute the unbiased pass@10 estimate to evaluate the functional correctness of the model's output, with a temperature coefficient of 0.8.

## C  Efficiency Analysis

In this section, we examine the efficiency of the proposed graph structure learning (GSL) module and influence maximization (IM) module. Specifically, we adopt the total running time and GPU memory usage as evaluation metrics. The results are reported in Table 3. As can be seen, our approach only spend around two hours for selecting 10% data from an instruction set containing 100,000 samples, which is efficient compared with instruction tuning stage, *e.g.*, fine-tuning LLaMA-2-7B on 100,000 instruction samples require over 12 hours in practice.

Received 20 February 2007; revised 12 March 2009; accepted 5 June 2009

