# OpenReview forum: "Automatic Instruction Data Selection for Large Language Models via Uncertainty-Aware Influence Maximization"
_ACM.org/TheWebConf/2025/Conference — WWW 2025 Poster_

### Official Review · Reviewer_hBQX · 2024-11-24

**Novelty:** 5
**Technical Quality:** 6

**Review:**

The paper proposes a data selection framework that explicitly incorporates the complex inter-dependencies within instruction data, called UniMax. The approach defines a latent instruction graph, and uses an uncertainty-aware influence function to score each example on it. Experiments validate the efficacy.

Pros:

1. The paper targets an important problem of LLMs - the potential bias and sub-optimal performance that may arise during the process of instruction tuning.

2. Data selection is an old and typical method. But it is somewhat novel that it is used to capture dependencies within instruction data.

3. The paper writing is generally clear.

4. The method performs well in experiments.

Cons:

To be honest, I am confused about ‘’the uncertainty score’’ in the Stage 1 of Figure 2. Why do we need pre-trained LLM to output uncertainty score in Stage 1? What is the correlation with the uncertainty score in the Stage 2?

**Questions:**

Please list questions for the authors for the discussion period, involving issues that an author response could change your opinion, clarify a confusion, or address a limitation.

Please address my comments listed in the previous section (Weaknesses). Some additional questions are as follows:

1. How confident are you that the graph structure learner accurately reflects the true dependencies within the data? Could you discuss potential biases introduced by this proxy?

2. What is the robustness of your approach to noisy or incorrect instruction examples? How might the presence of such data affect the graph structure and subsequent data selection?

**Reviewer Confidence:**

3: The reviewer is confident but not certain that the evaluation is correct

**Scope:**

4: The work is relevant to the Web and to the track, and is of broad interest to the community

---

### Official Review · Reviewer_NCmK · 2024-11-25

**Novelty:** 3
**Technical Quality:** 4

**Review:**

Pros
1: The paper reframes instruction data selection as an influence maximization problem, offering a new perspective and demonstrating innovation in methodology.
Comprehensive Methodology:
2: The paper provides proofs for properties like monotonicity and submodularity, giving a theoretical guarantee for the greedy algorithm used in selection.
3: Extensive experiments across multiple datasets (e.g., Flan-V2, CoT, and Code-Alpaca) and benchmarks (e.g., MMLU, TyDiQA, GSM, BBH, and Codex-Eval) confirm the effectiveness of the approach. Parameter sensitivity analysis and ablation studies provide valuable practical guidance for applying the method.

Cons:
1: The reliance on a proxy graph structure learner might introduce biases, as it may not perfectly capture relationships relevant to the target model.
2: The framework does not explicitly address the presence of noisy or incorrect instructions, which could degrade performance in real-world settings.
3: Experiments are conducted on carefully curated datasets, lacking scenarios involving noisy or diverse real-world instruction datasets.
4: The paper mentions but does not address ensuring consistency between the graph learner and the target LLM, which could lead to suboptimal data selection.

**Questions:**

1: Why does the paper use a linear attention mechanism instead of standard softmax attention for graph learning?
2: How does the proposed uncertainty-aware influence maximization ensure a balance between influence and uncertainty in data selection?
3: What are the theoretical guarantees provided for the greedy selection algorithm used in the framework?

**Ethics Review Flag:**

Yes

**Reviewer Confidence:**

3: The reviewer is confident but not certain that the evaluation is correct

**Scope:**

4: The work is relevant to the Web and to the track, and is of broad interest to the community

---

### Official Review · Reviewer_CkQH · 2024-11-26

**Novelty:** 5
**Technical Quality:** 5

**Review:**

The paper presents a novel framework, UniMax, for selecting instruction data that enhances the capabilities of Large Language Models (LLMs). The approach is grounded in graph influence maximization and incorporates a self-supervised graph learner to capture complex inter-dependencies within instruction data, Experimental results demonstrate the effectiveness of the proposed approach.

pros:

-	The proposal to use graph influence maximization for instruction data selection is innovative.
-	The paper provides a detailed introduction to the entire process from constructing a latent instruction graph to an uncertainty-aware influence function.
-	Extensive experiments on public datasets demonstrate the effectiveness of the approach.
-	The use of a self-supervised graph learner is a smart way to uncover latent structures without heavy computational overhead.

cons:

-	The paper could provide more details on the complexity of the graph learner and how it generalizes to different datasets.
-	The paper does not explicitly discuss the impact of noisy or incorrect instruction examples on the graph structure and data selection.
-	While the paper touches on hyperparameter sensitivity, a deeper analysis or guidelines for selecting these parameters could be beneficial.

**Questions:**

-	How does the proposed framework generalize to other LLMs beyond the ones tested? Are there any model-specific adjustments needed?
-	What is the robustness of the UniMax framework to noisy or incorrect instruction examples? Could you discuss potential strategies to improve robustness?
-	How confident are you in the consistency between the graph structure learner and the target model? Are there any plans to close this gap?
-	Could you provide more insights into how the hyperparameters, especially the influential path length and activation threshold, were chosen? Are there any guidelines for selecting these parameters for different datasets?
-	Have you considered any real-world applications where your method could be particularly beneficial? If so, what are the challenges and potential solutions?
-	How does your method compare with the state-of-the-art in terms of computational efficiency and performance? Are there any specific scenarios where your approach excels or falls short? Although this article is relatively new, I still hope to see a comparison between you and them.
Liu L, Liu X, Wong D F, et al. SelectIT: Selective Instruction Tuning for LLMs via Uncertainty-Aware Self-Reflection[C]//The Thirty-eighth Annual Conference on Neural Information Processing Systems.
-	In scenarios requiring long-term training or continual learning, how does the proposed method ensure stability and avoid potential biases?

**Reviewer Confidence:**

3: The reviewer is confident but not certain that the evaluation is correct

**Scope:**

3: The work is somewhat relevant to the Web and to the track, and is of narrow interest to a sub-community

---

### Official Review · Reviewer_PA7H · 2024-12-01

**Novelty:** 5
**Technical Quality:** 5

**Review:**

This paper studies the problem of instruction data selection for LLMs, pointing out the limitation of existing work in overlooking correlations between instruction examples. The paper proposes a framework UniMax that uses the dependencies within instruction data for LLMs, treating the problem via graph influence maximization by taking the target instruction dataset as a latent graph with nodes as instruction examples and edges as relations.

Pros:

1.	A new graph influence maximization perspective is involved in this paper for the studied problem.

2.	A self-learning mechanism is involved to mitigate the complexity issue, and an uncertain-aware method is embedded in the proposed framework.

3.	Extensive experiments on public datasets show that the proposed method outperforms baselines.

Cons:

1.	More experiments on the efficiency and scalability advanced by the self-learning mechanism can improve the paper.

2.	It would be beneficial if the variance over multiple runs of experiments could be provided.

3.	A deeper analysis of the algorithm performance with different models could be helpful.

**Questions:**

Could the authors share further experiments and related analyses for the efficiency/scalability of the proposed method?

**Reviewer Confidence:**

3: The reviewer is confident but not certain that the evaluation is correct

**Scope:**

4: The work is relevant to the Web and to the track, and is of broad interest to the community

---

### Official Review · Reviewer_h6No · 2024-12-01

**Novelty:** 4
**Technical Quality:** 4

**Review:**

The paper presents an Uncertainty-aware Influence Maximization (UniMax) framework for selecting high-value subsets of instruction data for large language models (LLMs). While the experiments demonstrate that the proposed approach often achieves comparable or superior performance using smaller subsets of data, the theoretical underpinning of the core assumption—that influence maximization consistently leads to an optimal subset—lacks direct proof. The experimental results can be more convincing with noisy or domain-specific data scenarios. The writing is mostly clear, with a well-structured presentation of the framework and experimental results. However, certain sections, such as the graph learner construction and the interplay of influence and uncertainty, require further elaboration.

Pros:
1. Introduction of UniMax to unify influence and uncertainty for data selection is new and grounded in a submodular optimization framework.
2. The linear complexity of the graph learner and scalability to large datasets are strong practical contributions.
3. Extensive comparisons across multiple datasets and benchmarks demonstrate the framework's effectiveness.

Cons:
1. The core assumption—that maximizing influence leads to an optimal subset—is not rigorously proven beyond indirect empirical validation.
2. The results can be more convincing with experiments conducted on some noisy or domain-specific datasets.
3. The proposed graph learner could introduce biases if latent relationships are misrepresented, which is not thoroughly analyzed.

**Questions:**

1. Core assumption validation: can the authors provide more rigorous theoretical evidence or experiments to validate that maximizing influence consistently leads to an optimal subset for instruction tuning?
2. Noisy data handling: how does the proposed framework handle noisy or mislabeled instruction data, and does performance degrade significantly in such scenarios?
3. Why is the chosen graph learner architecture appropriate for instruction data? Could alternative methods (e.g., similarity-based graphs) improve performance?
4. How does the framework scale computationally for datasets significantly larger than those evaluated?

**Reviewer Confidence:**

3: The reviewer is confident but not certain that the evaluation is correct

**Scope:**

3: The work is somewhat relevant to the Web and to the track, and is of narrow interest to a sub-community